# Targeted next-generation sequencing of bronchoalveolar lavage fluid for pathogen detection in connective tissue disease-related pulmonary infections

Xiaoyan Zhong,[1,2] Yuheng Xing,[1] Yuexi Zhang,[1] Zhiheng Luo,[3] Wu Xu,[3] Xu Chen,[1] Meng Liu,[1] Guanqun Yi,[1] Yukai Huang,[1] Shaoling Zheng,[1] Weiming Deng,[1] Xuechan Huang,[1] Tianwang Li,[1,4] Zhengping Huang[1]

ABSTRACT  The study aims to assess the clinical value of targeted next-generation sequencing of bronchoalveolar lavage fluid (BALF-tNGS) in pulmonary infection associated with connective tissue diseases (CTDs) and characterize the infection profiles in these patients. CTD patients undergoing bronchoscopy with BALF-tNGS between January 2022 to July 2025 were enrolled. tNGS results were compared with conventional microbiological tests (CMTs). Logistic regression identified risk factors for monomicrobial infection and polymicrobial infection. Among 124 CTD patients enrolled, 45 (36.3%) were male with the mean age of 52.6 ± 15.4 years, and the mean disease duration was 4.1 ± 6.1 years. CMTs detected pathogens in 43 patients, yielding 55 isolates: 23 bacteria, 13 viruses, 12 fungi, and 9 *Mycoplasma pneumoniae*. tNGS identified pathogens in 105 patients, yielding 168 isolates: 70 bacteria, 48 viruses, 43 fungi, and 7 *M. pneumoniae*. The sensitivity of tNGS was significantly higher than that of CMTs ($P = 0.02$). Treatment regimens were adjusted in 54 patients (49.5%) based on tNGS results. Polymicrobial infection was observed in 44 patients (40.37%), whereas 65 patients (59.63%) had monomicrobial infection. Multivariable analysis identified interstitial lung disease (ILD; odds ratio [OR] 3.90, 95% confidence interval [CI] 1.42–10.72), chronic kidney disease (CKD; OR 6.75, 95% CI 1.09–41.81), and decreased T-cell count (OR 0.30, 95% CI 0.13–0.69) as independent risk factors for polymicrobial infection. BALF-tNGS substantially increases the sensitivity of pathogen detection in CTD-related pulmonary infection and guides pathogen-directed antimicrobial therapy. Polymicrobial infection is common in CTD patients; ILD, CKD, and reduced T-cell counts are independent risk factors for its development.

**IMPORTANCE** The study delineates the infection landscape of connective tissue disease-related pneumonia and demonstrates that targeted next-generation sequencing of bronchoalveolar lavage fluid (BALF-tNGS) doubles the yield of identified pathogens, frequently revealing polymicrobial disease. This molecular insight prompted a change in antimicrobial therapy in nearly half of the cohort. Patients with concurrent interstitial lung disease, renal impairment, or lymphopenia were particularly susceptible to mixed infections. By enabling pathogen-directed treatment, BALF-tNGS reduces unnecessary antibiotic exposure, shortens recovery, and supports early, precision therapy.

KEYWORDS  targeted next-generation sequencing, bronchoalveolar lavage fluid, connective tissue disease, pulmonary infections, lupus

Connective tissue diseases (CTDs), including rheumatoid arthritis (RA), systemic lupus erythematosus (SLE), dermatomyositis (DM), and primary Sjögren's syndrome (pSS), refer to a cluster of autoimmune disorders characterized by chronic inflammation and

**Peer Reviewer** Shivaprakash M. Rudramurthy, Postgraduate Institute of Medical Education and Research, Chandigarh, India

Address correspondence to Zhengping Huang, zhuangrheu@gmail.com, or Tianwang Li, litianwang@163.com.

Xiaoyan Zhong, Yuheng Xing, and Yuexi Zhang contributed equally to this article. The author order was determined based on the sequence of their contributions to the core research stages.

The authors declare no conflict of interest.

damage to connective tissues, affecting multiple organ systems including the skin, joints, blood vessels, and internal organs (1–5). These diseases arise from immune dysregulation, leading to autoantibody production and systemic manifestations (1, 6).

In nearly all CTDs, respiratory events of varying acuity can occur—ranging from subacute to acute, hyperacute, and even catastrophic or fulminant (7). These may manifest as systemic, life-threatening conditions that progress rapidly to respiratory failure, requiring mechanical ventilation and multiple targeted therapies, and ultimately serve as the terminal event for most affected patients. Acute respiratory events in CTDs can involve every anatomical component of the respiratory system, including the airways, lung parenchyma, alveolar capillaries, pulmonary vessels, pleura, and ventilatory muscles. These events may also induce extrapulmonary complications specific to the disease, such as gastrointestinal or cardiac pathologies, and further trigger-related acute respiratory events. For instance, the unique pathophysiological mechanisms in patients with scleroderma and dermatomyositis-polymyositis (DM-PM) can severely impair swallowing function, leading to aspiration pneumonia. Meanwhile, chronic inflammation associated with CTDs contributes to both atherosclerotic and non-atherosclerotic processes, which can subsequently result in acute respiratory events like pericarditis-induced dyspnea (8). CTDs are frequently complicated by severe pulmonary infections that are the leading cause of acute exacerbation, prolonged hospitalization, admission to the ICU, and death (9). The mortality may exceed 50% in patients admitted for infection (9). Unfortunately, conventional microbiological tests (CMTs) exhibit unacceptably low sensitivity (e.g., ≤30% for *Pneumocystis jirovecii*), leading to frequent missed diagnoses and delayed targeted therapy. A retrospective study analyzed 192 PM/DM patients from a tertiary medical center (1999–2008). Major infections occurred in 27.6% (53/192) of patients, with an incidence rate of 11.1 episodes per 100 patient-years; 7.8% experienced recurrent infections. The primary pathogens causing infections in PM/DM patients include *Staphylococcus aureus*, *Klebsiella*, *Escherichia coli*, *Mycobacterium*, and *Salmonella*. The most common infections were aspiration pneumonia (21.1%) and opportunistic infections (18.4%). Among the opportunistic infections, *Mycobacterium* was the most prevalent, accounting for half of all opportunistic infection episodes, followed by *cytomegalovirus* (CMV) infection (10).

CMTs, including sputum, blood, or bronchoalveolar lavage fluid (BALF) cultures and PCR, are widely used for diagnosing lung infections (11–13). However, their low sensitivity—particularly for fastidious or intracellular pathogens—often yields false-negative results, delaying appropriate treatment. In CTD-related pulmonary infections, the pathogen spectrum is broad, encompassing bacteria (e.g., *Pseudomonas aeruginosa* and *Klebsiella pneumoniae*), fungi (e.g., *Pneumocystis jirovecii*), viruses (e.g., CMV), and rare pathogens, often as polymicrobial co-infections. CTD itself can cause pneumonic changes, while immunosuppressive therapy further blunts the typical clinical and radiographic features of infection, diminishing the utility of chest radiography or CT. Consequently, empirical broad-spectrum antibiotics are overprescribed, accelerating antimicrobial resistance without improving outcomes. Thus, traditional approaches are insufficient for rapid diagnosis and treatment of pulmonary infections in CTDs, and delays are common.

Recent advances in molecular diagnostics, particularly targeted next-generation sequencing (tNGS), offer a promising solution by enabling comprehensive and rapid pathogen detection. BALF is an optimal specimen for pulmonary infection diagnosis due to its direct sampling from the lower respiratory tract. BALF-tNGS has demonstrated superior sensitivity compared to CMTs in immunocompromised populations, including CTD patients. However, the clinical utility of BALF-tNGS in CTD-associated pulmonary infections remains underexplored, particularly regarding pathogen profiles and diagnostic performance.

This study aims to evaluate the pathogen detection rate and microbial distribution in lower respiratory tract infections among CTD patients by comparing tNGS technology with CMTs. The research will assess the advantages and disadvantages of

**TABLE 1** General characteristics of the patients

| Variable | Total ($n = 124$) |
|---|---|
| Male, $n$ (%) | 45 (36.3) |
| Age, years | 52.6 ± 15.4 |
| Disease duration, years | 4.1 ± 6.1 |
| Smoking history, $n$ (%) | 26 (21.0) |
| ILD, $n$ (%) | 54 (43.5) |
| Diabetes, $n$ (%) | 16 (12.9) |
| Hypertension, $n$ (%) | 29 (23.4) |
| CKD, $n$ (%) | 12 (9.7) |
| Hypoproteinemia, $n$ (%) | 66 (53.2) |
| WBC (×$10^9$/L) | 8.8 ± 4.8 |
| Neutrophils (×$10^9$/L) | 6.8 ± 4.5 |
| T cells (×$10^9$/L) | 1.3 ± 0.8 |
| CRP, mg/L | 44.1 ± 61.3 |
| ESR, mm/1st h | 59.3 ± 37.9 |
| RA, $n$ (%) | 26 (21.0) |
| SLE, $n$ (%) | 25 (20.2) |
| DM, $n$ (%) | 17 (13.7) |
| pSS, $n$ (%) | 13 (10.5) |
| AAV, $n$ (%) | 11 (8.9) |
| PM, $n$ (%) | 10 (8.1) |
| ASS, $n$ (%) | 6 (4.8) |
| UCTD, $n$ (%) | 5 (4.0) |
| SD, $n$ (%) | 4 (3.2) |
| SSc, $n$ (%) | 4 (3.2) |
| BD, $n$ (%) | 2 (1.6) |
| OS, $n$ (%) | 1 (0.8) |

next-generation sequencing (NGS) versus traditional pathogen detection techniques, while investigating high-risk factors for polymicrobial infections in CTD patients. The findings are expected to provide evidence for early etiological diagnosis and precise clinical treatment.

## RESULTS

### General characteristics

A total of 124 patients who were admitted to the Department of Rheumatology and Immunology were enrolled according to the strict enrollment criteria in this retrospective study (Table 1). Out of all patients, 45 (36.3%) were male, and the average age was 52.6 ± 15.4 years. The mean disease duration was 4.1 ± 6.1 years. Of the patients, 26 (21.0%) had a smoking history, 54 (43.5%) had interstitial lung disease (ILD), 16 (12.9%) had diabetes mellitus, 29 (23.4%) had hypertension, 12 (9.7%) had chronic kidney disease (CKD), and 66 (53.2%) had hypoalbuminemia. Laboratory parameters were as follows: white blood cell count 8.8 ± 4.8 × $10^9$/L, neutrophil count 6.8 ± 4.5 × $10^9$/L, T-cell count 1.3 ± 0.8 × $10^9$/L, C-reactive protein (CRP) 44.1 ± 61.3 mg/L, and erythrocyte sedimentation rate (ESR) 59.3 ± 37.9 mm/h. Twelve distinct CTDs were included (Fig. 1; Table S1): RA (26 cases, 21.0%), SLE (25 cases, 20.2%), DM (17 cases, 13.7%), pSS (13 cases, 10.5%), anti-neutrophil cytoplasmic antibody-associated vasculitis (AAV; 11 cases, 8.9%), PM (10 cases, 8.1%), anti-synthetase syndrome (ASS; 6 cases, 4.8%), undifferentiated connective-tissue disease (UCTD; 5 cases, 4.0%), Still's disease (SD; 4 cases, 3.2%), systemic sclerosis (SSc; 4 cases, 3.2%), Behçet's disease (BD; 2 cases, 1.6%), and overlap syndrome (OS, 1 case, 0.8%).

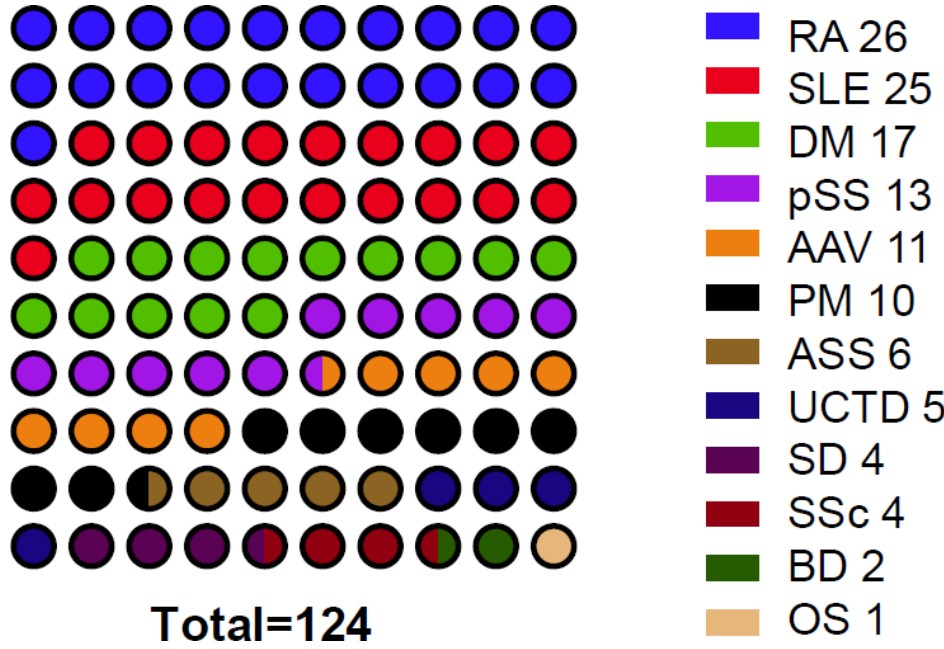

**FIG 1** Dot plot representation of CTD cases.

## Pathogens detected by CMTs

With the application of CMTs, 43 patients were positive for ≥1 pathogen, yielding 55 discrete organisms in total (Fig. 2; Table S2). Bacteria accounted for the largest proportion ($n$ = 23, 41.8%), followed by viruses ($n$ = 13, 23.6%), fungi ($n$ = 12, 21.8%), and *Mycoplasma pneumoniae* ($n$ = 7, 12.7%). The leading representatives within each category were *Klebsiella pneumoniae* ($n$ = 8, 14.6%), EBV ($n$ = 6, 10.9%), *Candida albicans* ($n$ = 6, 10.9%), and *M. pneumoniae* ($n$ = 7, 12.7%), respectively.

## Pathogens detected by tNGS

Using tNGS, we detected a higher number of positive patients than with conventional methods (105 vs 43; Fig. 3; Table S3). Among the 168 pathogens identified, bacteria were most frequent (70, 41.7%), followed by viruses (48, 28.6%) and fungi (43, 25.6%); *Mycoplasma pneumoniae* accounted for the remaining 7 cases (4.2%). Among the bacterial cohort, *Haemophilus influenzae* led the tally with 15 cases (8.9%), followed closely by *Klebsiella pneumoniae* at 11 cases (6.5%) and *Pseudomonas aeruginosa* at 8 cases (4.8%), together sketching the predominance of Gram-negative organisms. On the

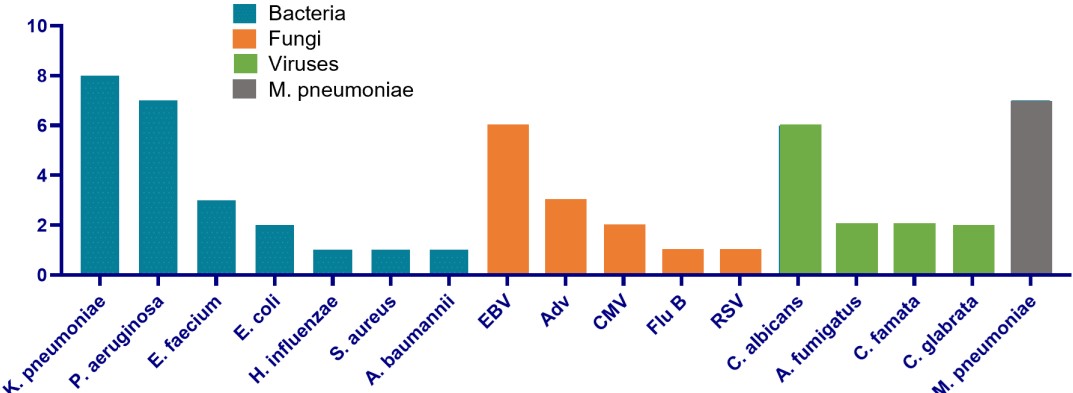

**FIG 2** Detection results of pathogens by traditional methods.

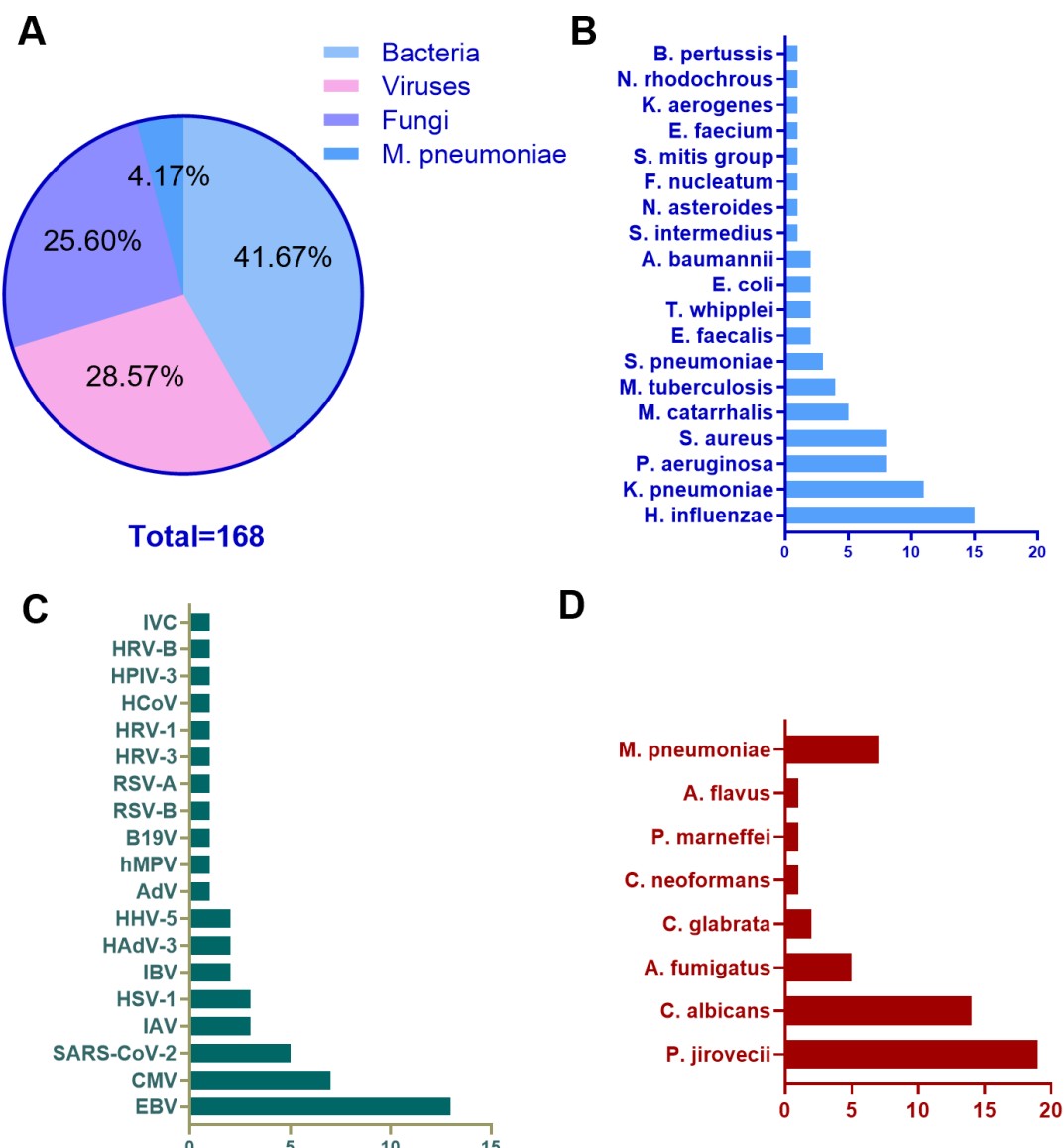

**FIG 3** Detection results of pathogens by tNGS. (A) Proportion of pathogens; (B) bar chart of bacterial distribution; (C) bar chart of viral distribution; (D) bar chart of fungi and mycoplasma distribution.

viral front, EBV proved most active, accounting for 13 cases (7.7%), while CMV (7 cases, 4.2%) and *severe acute respiratory syndrome coronavirus 2* (5 cases, 2.9%) came next, underscoring the coexistence of latent and emerging infections. Fungi were dominated by *Pneumocystis jirovecii* with 19 cases (11.3%), trailed by *Candida albicans* (14 cases, 8.3%), and *Aspergillus fumigatus* (5 cases, 3.0%), highlighting the opportunistic resurgence in immunocompromised hosts. Finally, *Mycoplasma pneumoniae* stood in a class of its own, detected in seven cases (4.2%), adding a note of "atypical" etiology to the pathogen landscape.

## Comparison of CMTs and tNGS

Among 124 specimens, tNGS identified pathogens in 105 (84.68%), whereas CMTs were positive in only 42 (33.87%) (Table 2). After integrating clinical, radiological, and therapeutic data, 109 cases were adjudicated as true infections. Relative to this reference, tNGS demonstrated a sensitivity of 96.33% (105/109), markedly superior to the 38.53%

**TABLE 2** Positive rate and sensitivity of CMTs and tNGS

| Method | Positive rate | Sensitivity |
|---|---|---|
| CMTs | 33.87% | 38.53% |
| tNGS | 84.68% | 96.33% |

(42/109) achieved by CMTs. Chi-square test confirmed that this difference was statistically significant ($x^2 = 6.62$, $P = 0.02$) (Table 3).

## The influence of tNGS on anti-infective regimen

Of 124 patients, 66 (53.2%) had their antimicrobial therapy modified after tNGS results, whereas 43 (34.7%) continued the original regimen because it remained effective. Among the 66 modifications, drugs were added in 38 (30.6%), switched in 24 (19.4%), and discontinued in 4 (3.2%). Sulfamethoxazole was the most commonly adjusted agent (19 patients, 15.3%). Clinical improvement was achieved in 119 patients (96.0%); the remaining 5 (4.0%) were transferred to the ICU for worsening infection.

## Polymicrobial pulmonary infections and their risk factors

Of the 109 confirmed infections, 44 had polymicrobial infections (Table 4). Among them, 29 cases (65.9%) had infections with two types of pathogens, 14 cases (31.8%) had infections with three types of pathogens, and 1 case (2.3%) had an infection with four types of pathogens. In univariate binary logistic regression analysis (Table 5), polymicrobial pulmonary infections were associated with gender, disease duration, smoking history, ILD, CKD, hypoproteinemia, and T-cell counts. In the multivariable model (Table 6), only three predictor variables were identified through multivariable regression analysis: ILD (odds ratio [OR] = 3.90, 95% confidence interval [CI] = 1.415–10.719, $P = 0.008$), CKD (OR = 6.753, 95% CI = 1.091–41.808, $P = 0.04$), and T-cell counts (OR = 0.300, 95% CI = 0.133–0.678, $P = 0.004$). The results showed that polymicrobial infections were positively correlated with ILD and CKD and negatively correlated with T-cell counts.

## DISCUSSION

Pulmonary infections remain a critical challenge in patients with CTDs, contributing significantly to morbidity and mortality due to the complex interplay of immune dysregulation, immunosuppressive therapy, and overlapping disease-related organ damage. In this study, we evaluated the clinical utility of BALF-tNGS for pathogen detection in CTD-related pulmonary infections, compared its performance with CMTs, and identified risk factors for polymicrobial infections. Our findings highlight the superiority of BALF-tNGS in enhancing diagnostic sensitivity, characterizing the dominant pathogen spectrum, and underscore the clinical relevance of identifying predictors of polymicrobial disease.

A key finding of our study is the substantially higher sensitivity of BALF-tNGS (96.33%) compared to CMTs (38.53%) in detecting pathogens among confirmed CTD-related pulmonary infections. This aligns with growing evidence that NGS technologies outperform traditional methods in immunocompromised populations, where subtle clinical manifestations and fastidious or rare pathogens often lead to underdiagnosis (14, 15). CTD patients, frequently treated with high-dose glucocorticoids, immunosuppressants, or biologic agents, are particularly vulnerable to atypical or opportunistic

**TABLE 3** Chi-square test for CMTs and tNGS

| tNGS | CMTs | | | $x^2$ | P value |
|---|---|---|---|---|---|
| | Positive | Negative | Total | | |
| Positive | 38 | 67 | 105 | 6.62 | 0.02 |
| Negative | 4 | 0 | 4 | | |
| Total | 42 | 63 | 109 | | |

**TABLE 4** Comparison of clinical characteristics between patients with monomicrobial infection and those with polymicrobial infection

| Variable | Polymicrobial (n = 44) | Monomicrobial (n = 65) | P value |
|---|---|---|---|
| Male, n (%) | 24 (54.5) | 18 (40.9) | 0.0047 |
| Age, years | 55.9 ± 16.4 | 50.4 ± 14.4 | 0.062 |
| Disease duration, years | 2.4 ± 4.1 | 5.4 ± 7.3 | 0.008 |
| Smoking history, n (%) | 16 (36.4) | 8 (18.2) | 0.0029 |
| ILD, n (%) | 26 (59.1) | 19 (43.2) | 0.0028 |
| Diabetes, n (%) | 6 (13.6) | 7 (15.9) | 0.65 |
| Hypertension, n (%) | 11 (25.0) | 14 (31.8) | 0.24 |
| CKD, n (%) | 8 (18.2) | 3 (6.8) | 0.027 |
| Hypoproteinemia, n (%) | 30 (68.2) | 28 (63.6) | 0.01 |
| WBC ($\times 10^9$/L) | 8.7 ± 5.8 | 8.5 ± 4.1 | 0.57 |
| Neutrophils ($\times 10^9$/L) | 7.2 ± 5.5 | 6.3 ± 3.7 | 0.69 |
| T cells ($\times 10^9$/L) | 0.87 ± 0.56 | 1.5 ± 0.80 | <0.0001 |
| CRP, mg/L | 48.0 ± 59.6 | 41.1 ± 58.1 | 0.37 |
| ESR, mm/first h | 62.8 ± 34.8 | 60.1 ± 41.6 | 0.58 |
| RA, n (%) | 6 (13.6) | 16 (24.6) | 0.16 |
| SLE, n (%) | 6 (13.6) | 14 (21.5) | 0.3 |
| DM, n (%) | 9 (20.5) | 7 (10.8) | 0.16 |
| pSS, n (%) | 5 (11.4) | 7 (10.8) | 0.92 |
| AAV, n (%) | 6 (13.6) | 5 (7.7) | 0.35 |
| PM, n (%) | 3 (6.8) | 5 (7.7) | >0.99 |
| ASS, n (%) | 3 (6.8) | 3 (4.6) | 0.68 |
| UCTD, n (%) | 2 (4.6) | 3 (4.6) | >0.99 |
| AOSD, n (%) | 3 (6.8) | 1 (1.5) | 0.3 |
| SSc, n (%) | 1 (2.3) | 1 (1.5) | >0.99 |
| BD, n (%) | 0 | 2 (3.1) | 0.51 |
| OS, n (%) | 0 | 1 (1.5) | >0.99 |

infections, which CMTs—reliant on culture, antigen detection, or limited PCR panels—often miss (9, 16, 17). For instance, *Pneumocystis jirovecii*, a major fungal pathogen in our cohort, is notoriously difficult to detect via CMTs, and tNGS identified 19 cases, highlighting its utility in capturing such elusive pathogens.

The clinical impact of this improved sensitivity is underscored by the fact that 53.2% of patients had their anti-infective regimens adjusted based on tNGS results, with modifications including targeted drug additions, switches, or discontinuations. This suggests that BALF-tNGS can move clinical practice beyond empiric broad-spectrum therapy—often necessitated by diagnostic uncertainty in CTDs—to pathogen-directed interventions, potentially reducing unnecessary antimicrobial exposure and mitigating the risk of drug resistance (15). Notably, sulfamethoxazole, a first-line agent for *P. jirovecii* and other opportunistic infections, was the most frequently adjusted drug, further supporting the role of tNGS in identifying pathogens that guide specific therapeutic choices.

Our analysis revealed distinct pathogen profiles in CTD patients: *Haemophilus influenzae* (bacterial), EBV (viral), and *Pneumocystis jirovecii* (fungal) emerged as the most prevalent pathogens in their respective categories. These findings reflect the unique immunological landscape of CTDs. *H. influenzae*, a common cause of respiratory infections in immunocompromised hosts, thrives in the setting of impaired mucosal immunity secondary to CTDs or their treatments (18). EBV, a latent herpesvirus, may reactivate in the context of T-cell dysfunction—consistent with our observation of reduced T-cell counts in patients with polymicrobial infections—leading to persistent or opportunistic viral pathogenesis (19).

The prominence of *P. jirovecii* (11.3% of all pathogens) is particularly noteworthy. This fungus is a well-recognized threat in patients receiving long-term immunosuppression,

TABLE 5 Associations between single infection and co-infections

| Variable | β | OR (95% CI) | P value |
| --- | --- | --- | --- |
| Age | 0.024 | 1.025 (0.998–1.052) | 0.072 |
| Gender | 1.142 | 3.133 (1.402–7.005) | 0.005 |
| Disease duration | −0.112 | 0.894 (0.812–0.986) | 0.024 |
| Smoke | 1.404 | 4.071 (1.556–10.651) | 0.004 |
| ILD | 1.252 | 3.497 (1.565–7.816) | 0.002 |
| CKD | 1.524 | 4.593 (1.145–18.420) | 0.031 |
| Hypoproteinemia | 1.041 | 2.832 (1.270–6.316) | 0.011 |
| Tcell | −1.456 | 0.233 (0.113–0.479) | <0.001 |
| Diabetes | 0.269 | 1.308 (0.408–4.193) | 0.651 |
| Hypertension | 0.194 | 1.214 (0.492–2.996) | 0.673 |
| WBC | 0.008 | 1.008 (0.931–1.091) | 0.848 |
| Neu | 0.046 | 1.047 (0.962–1.140) | 0.285 |
| CRP | 0.002 | 1.002 (0.995–1.009) | 0.553 |
| ESR | 0.002 | 1.002 (0.992–1.012) | 0.727 |

including CTD patients on high-dose glucocorticoids or rituximab (20–22). The high detection rate via tNGS, compared to CMTs (which identified only 6 cases of *Candida albicans* as the dominant fungus), emphasizes the limitations of traditional methods in identifying this pathogen and reinforces the value of tNGS in guiding prophylaxis and treatment for at-risk CTD subgroups, such as those with ILD or lymphopenia. However, some ORs (e.g., for CKD) are accompanied by wide CIs, primarily due to the small subgroup sample size and the low number of events, resulting in imprecise estimates. Therefore, interpretations of the true effect size of CKD should be made cautiously, and no definitive quantitative risk estimate can be drawn from these data.

Polymicrobial infections were observed in 40.37% of confirmed cases, with independent risk factors identified as ILD, CKD, and reduced T-cell counts. These findings align with the pathophysiological basis of CTD-related immune dysfunction and organ damage. ILD, a common manifestation in CTDs such as DM and SSC, disrupts pulmonary architecture and mucosal barrier function, creating a permissive environment for multiple pathogens (23). CKD, often complicating CTDs like SLE, impairs both innate and adaptive immunity—via uremic toxin accumulation, reduced antibody production, and neutrophil dysfunction—predisposing to recurrent or polymicrobial infections (24).

Notably, reduced T-cell counts emerged as a strong inverse predictor of polymicrobial disease (OR = 0.300), which is consistent with the critical role of T cells in controlling intracellular pathogens (e.g., viruses, *Mycobacterium*) and coordinating antimicrobial immunity (25, 26). This finding underscores the importance of monitoring T-cell subsets in CTD patients, as lymphopenia may serve as a biomarker to identify those at highest risk of complicated infections, warranting closer surveillance and preemptive interventions.

Our study has several limitations. First, it is a single-center retrospective analysis, which may limit generalizability due to potential biases in patient selection and local pathogen epidemiology. Future research should aim to validate these findings through multicenter, prospective cohort designs and include broader patient populations to

TABLE 6 Analyses of risk factors associated with co-infections

| Variable | β | OR (95% CI) | P value |
| --- | --- | --- | --- |
| Gender | 0.511 | 1.667 (0.4665–5.978) | 0.433 |
| Disease duration | −0.75 | 0.927 (0.829–1.037) | 0.187 |
| Smoke | 1.205 | 3.337 (0.722–15.433) | 0.123 |
| ILD | 1.36 | 3.895 (1.415–10.719) | 0.008 |
| CKD | 1.91 | 6.753 (1.091–41.808) | 0.04 |
| Hypoproteinemia | 0.85 | 2.340 (0.817–6.702) | 0.113 |
| T-cell counts | −1.202 | 0.300 (0.133–0.678) | 0.004 |

enhance external validity. Second, the interpretation of tNGS results relies on a composite of technical, etiological, and clinical criteria, as no standardized thresholds for "positivity" exist for respiratory pathogens in this context; this subjectivity may introduce variability. In our study, we interpreted tNGS results—whether positive or negative—in conjunction with the patient's clinical condition, other laboratory findings, and the pathogen's inherent pathogenicity to determine if it is the causative microorganism. When necessary, a comprehensive interpretation incorporating literature review or a multidisciplinary consultation involving infectious disease specialists, pathogenic microbiology teams, and relevant clinical departments was conducted. Third, due to the retrospective nature of our study and the relatively small sample size within each specific immunosuppressive regimen subgroup (e.g., only a small number of patients received biologic agents), a robust statistical comparison of infection risks across different regimens was not feasible. We did not systematically assess the impact of specific immunosuppressive agents (e.g., rituximab and cyclophosphamide) on infection risk, which warrants further investigation. Literature indicates that immunosuppressive therapy significantly increases the risk of infection. In patients with rheumatic diseases receiving immunosuppressants (such as rituximab, belimumab, and traditional immunosuppressive agents), the risk of severe infection is less associated with the specific type of drug and more dependent on the patient's underlying condition. Immunosuppressive therapy for rheumatic diseases significantly alters the spectrum of pathogens, predominantly leading to opportunistic infections (e.g., PJP and tuberculosis) and respiratory pathogens (e.g., pneumococcus and viruses). The distinct immunodeficiency profiles of different rheumatic diseases (such as SLE and RA) may result in specific pathogen susceptibilities; for instance, SLE patients are more susceptible to tuberculosis infections (27, 28). Finally, the study primarily assessed the short-term impact of tNGS on empirical antimicrobial therapy adjustment. Long-term outcomes (e.g., recurrence rates and mortality) were not evaluated, limiting our ability to correlate tNGS-guided therapy with sustained clinical benefit. Future multicenter prospective studies are needed to validate these findings and establish standardized protocols for tNGS implementation in CTD care.

## Conclusions

In summary, our study demonstrates that BALF-tNGS significantly enhances pathogen detection sensitivity in CTD-related pulmonary infections, outperforming CMTs in identifying both common and opportunistic pathogens. The dominant pathogens— *H. influenzae*, EBV, and *P. jirovecii*—reflect the unique immunological vulnerabilities of CTD patients, while the high rate of polymicrobial infections underscores the need for comprehensive diagnostic approaches. ILD, CKD, and reduced T-cell counts emerge as key risk factors for polymicrobial pulmonary infections, enabling targeted risk stratification.

These findings support the integration of BALF-tNGS into routine clinical practice for CTD patients with suspected pulmonary infections, particularly those with severe immunocompromised or refractory disease. By enabling rapid pathogen detection, BALF-tNGS has the potential to improve outcomes, reduce antimicrobial misuse, and mitigate the burden of infections in this high-risk population. Future multicenter prospective studies are needed to validate these findings and establish standardized protocols for tNGS implementation in CTD care.

## MATERIALS AND METHODS

### Human subjects

Patients with CTD were eligible if they were managed in the outpatient clinic or inpatient ward of the Department of Rheumatology and Immunology at Guangdong Second Provincial General Hospital between October 2022 and May 2025, underwent

fiber-optic bronchoscopy with bronchoalveolar lavage (BAL), and had BAL fluid submitted for tNGS.

## Ethical statement and informed consent

Research was granted ethical approval by the Ethics Committee of Guangdong Second Provincial General Hospital under the protocol number 2024-KY-KZ-285-01. The patients have been fully informed about the bronchoscopy procedure, the associated risks of collecting BALF, as well as potential sequencing results and any subsequent treatment plans. Informed consent was obtained from all participants prior to the collection of research samples. Clinical data were gathered through a retrospective review of medical records.

## Inclusion criteria

- Definitive diagnosis of a CTD (e.g., RA, SLE, DM, PM, ASS, Sjögren's syndrome, ANCA-associated vasculitis, UCTD, adult-onset SD, SSc, BD, or OS) according to established classification criteria.
- Presence of fever or respiratory symptoms (cough, expectoration, dyspnea, increased respiratory rate, etc.) or reduced percutaneous oxygen saturation, with chest CT or X-ray findings suggestive of—or not excluding—pulmonary infection.
- Voluntary consent to undergo fiber-optic bronchoscopy with BAL and to have the lavage fluid analyzed by high-throughput NGS for pathogen detection.

## Exclusion criteria

- Uncontrolled systemic infection.
- Relative contraindications to fiber-optic bronchoscopy.
- Inadequate or unsuitable specimen collection.
- Incomplete clinical data.

## Clinical data collection and specimen acquisition

Consecutive CTD patients who underwent bronchoscopy with BALF sent for tNGS were enrolled. The following data were extracted: diagnosis, age, sex, disease duration, smoking history, and histories of hypertension and diabetes. Upon admission, vital signs were monitored, and baseline laboratory tests—complete blood count, comprehensive metabolic panel, procalcitonin, ESR, CRP, and chest CT—were performed. Before the first dose of antibiotics, sputum was collected by natural expectoration, sputum catheter aspiration, and bronchoscopic collection and sent for bacterial culture, Gram stain, and a nine-pathogen respiratory panel (*Influenza A virus*, *Influenza B virus*, *Legionella*, *Mycoplasma pneumoniae*, *Chlamydophila pneumoniae*, *Adenovirus*, *Respiratory syncytial virus*, *Parainfluenza virus*, and *Rickettsia*). After obtaining informed consent, fiber-optic bronchoscopy with BAL was performed (targeting lesions identified on chest CT); the BALF was submitted for sputum smear, bacterial culture, (1,3)-β-D-glucan (BDG) test, galactomannan (GM) test, and tNGS. The GM and BDG assays were performed using the Fungal (1-3)-β-D-Glucan Detection Kit (Chromogenic Method) and the Aspergillus Galactomannan Detection Kit (Chemiluminescence Method) from China's Xinuo Company.

## Interpretation of tNGS results

At present, there is no unified industry threshold for determining "positive" results in tNGS reports for respiratory pathogens. Laboratories and clinical practices generally follow the "three-step comprehensive judgment" as follows.

## Technical positivity

- Detection limit: sequences with ≥100 copies/mL are reliably detected.
- Reporting symbol: a "(+)" marked in the result column indicates technical positivity.

## Etiological verification

- Common pathogens: need to be re-verified using *in vitro* diagnostic kits for the same type of specimen or Sanger sequencing (first-generation sequencing).
- Uncommon/rare pathogens: if routine tests are already positive, they can be directly recognized; otherwise, literature evidence supporting their pulmonary pathogenicity is required.

## Clinical judgment

A result can be finally confirmed as clinically positive only if at least one of the following conditions is met:

- consistent with the patient's symptoms, imaging findings, and treatment response;
- negative by traditional methods, but targeted medication is effective, or there is sufficient literature supporting its pathogenicity;
- consistent with traditional etiological results (culture, antigen/nucleic acid testing, etc.).

Whether a pathogen can be identified as the responsible one must be finally determined by clinicians based on the above composite criteria.

## Statistical analysis

Continuous variables were reported as mean ± standard deviation, whereas categorical variables were presented as frequencies and percentages. Group comparisons were performed using Student's *t*-test or Mann-Whitney *U* test for continuous variables and chi-square tests for categorical variables. Furthermore, patients were stratified into two groups: those with polymicrobial infection and those with monomicrobial infection. Univariate binary logistic regression analysis was conducted to screen potential risk factors associated with polymicrobial infection (significance threshold set at $P \leq 0.05$). Variables meeting this criterion were subsequently included in a multivariable regression model to identify independent predictors of polymicrobial infection, with statistical significance defined as $P < 0.05$ (enter method). The data were analyzed using SPSS software (version 23; IBM Corp., Armonk, NY, USA) and GraphPad Prism software (GraphPad Software, La Jolla, CA, USA).

### ACKNOWLEDGMENTS

Z.H. received funding from the National Natural Science Foundation of China (No. 82302025), Young Talent Support Project of Guangzhou Association for Science and Technology (No. QT-2024-032) and Guangzhou Science and Technology Plan Projects (No. 2024A03J0773). T.L. received funding from Guangzhou Science and Technology Plan Projects (No. 202102080321), Guangdong Yiyang Healthcare Charity Foundation (JZ2022001-1), and Natural Science Foundation of Guangdong Province (2024A1515013173).

X.Z.: methodology and writing—review and editing. Y.Z.: investigation, methodology, resources, and writing—review and editing. Y.X.: data curation, formal analysis, resources, and writing—original draft. Z.L. and W.X.: data curation. T.L.: methodology; writing—review and editing; funding acquisition, investigation, and supervision. Z.H.: conceptualization; data curation; formal analysis; funding acquisition; investigation; methodology;

project administration; resources; software; validation; visualization; writing—original draft; and writing—review and editing.

The authors declare that the research was conducted in the absence of any commercial or financial relationships that could be construed as a potential conflict of interest.

## AUTHOR AFFILIATIONS

¹Department of Rheumatology and Immunology, The Affiliated Guangdong Second Provincial General Hospital of Jinan University, Guangzhou, China
²Department of Internet Medical Center, The Affiliated Guangdong Second Provincial General Hospital of Jinan University, Guangzhou, China
³Department of Information Technology, The Affiliated Guangdong Second Provincial General Hospital of Jinan University, Guangzhou, China
⁴Department of Rheumatology and Immunology, Zhaoqing Central People's Hospital, Zhaoqing, China

## AUTHOR ORCIDs

Tianwang Li http://orcid.org/0000-0001-6438-2691
Zhengping Huang http://orcid.org/0000-0002-2955-7385

## DATA AVAILABILITY

The raw data supporting the conclusions of this article will be made available by the authors, without undue reservation.

## ETHICS APPROVAL

The study was approved by the Ethics Committee of Guangdong Second Provincial General Hospital (approval No.: 2023-KY-KZ-215-01). All procedures in the studies were performed in compliance with the local legislative requirements and institutional ethical guidelines. Written informed consent was obtained from all participants prior to their enrollment in the study.

## ADDITIONAL FILES

The following material is available online.

### Supplemental Material

**Supplemental tables (Spectrum02550-25-s0001.docx).** Tables S1 to S3.

### Open Peer Review

**PEER REVIEW HISTORY (review-history.pdf).** An accounting of the reviewer comments and feedback.

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
