## [Reviewer comments · Microbiology Spectrum]

Microbiology Spectrum

Targeted Next-Generation Sequencing of Bronchoalveolar Lavage Fluid for Pathogen Detection in Connective Tissue Disease–Related Pulmonary Infections

Xiaoyan Zhong, Yuheng Xing, Yuexi Zhang, Zhiheng Luo, Wu Xu, Xu Chen, Meng Liu, Guanqun Yi, Yukai Huang, Shaoling Zheng, Weiming Deng, Xuechan Huang, Tianwang Li, and Zhengping Huang

Corresponding Author(s): Zhengping Huang, Guangdong Second Provincial General Hospital

Review Timeline:

Submission Date:	August 19, 2025
Editorial Decision:	September 24, 2025
Revision Received:	October 24, 2025
Accepted:	October 27, 2025

Editor: Benjamin Liu

Reviewer(s): Disclosure of reviewer identity is with reference to reviewer comments included in decision letter(s). The following individuals involved in review of your submission have agreed to reveal their identity: Shivaprakash M Rudramurthy (Reviewer #2)

Transaction Report:

DOI: <https://doi.org/10.1128/spectrum.02550-25>

Re: Spectrum02550-25 (**Targeted Next-Generation Sequencing of Bronchoalveolar Lavage Fluid for Pathogen Detection in Connective Tissue Disease-Related Pulmonary Infections**)

Dear Prof. Zhengping Huang:

Thank you for the privilege of reviewing your work. Below you will find my comments, instructions from the Spectrum editorial office, and the reviewer comments.

Editor's comments:

Lines 89-90: "CMTs, including sputum, blood, or bronchoalveolar lavage fluid (BALF) cultures and PCR, are widely used for diagnosing lung infections": There are no references to support these statements. More references should be cited, with these (PMID: 39744807; PMID: 40137747; PMID: 33069878) as examples (citing is optional).

Please return the manuscript within 30 days; if you cannot complete the modification within this time period, please contact me. If you do not wish to modify the manuscript and prefer to submit it to another journal, notify me immediately so that the manuscript may be formally withdrawn from consideration by Spectrum.

Revision Guidelines

Sincerely,
Benjamin Liu
Editor
Microbiology Spectrum

Reviewer #2 (Comments for the Author):

The manuscript is vague and difficult to follow. The study design lacks clarity, with poorly defined cohorts and the absence of appropriate controls. It is also unclear how and when the organisms were classified as pathogenic. The authors should specify whether a cut-off for tNGS was defined to consider an organism pathogenic. Furthermore, the platform used for tNGS has not been mentioned, which is an important methodological detail that must be included.

Specific comments requiring clarification:

- Lines 71-74: Is aspiration pneumonitis truly an extra-pulmonary manifestation? How is it linked to the digestive tract and the heart?
- Lines 77-80: Complications appear to be repeated.
- Line 82: The mention of culture and serodiagnostics for *Pneumocystis jirovecii* is inappropriate, as this organism is not a good example for culture-based methods.
- Line 87: Please clarify whether these are pulmonary opportunistic infections.
- Line 138: *Candida albicans* should not be considered pathogenic flora in BALF.
- Line 166: While only 109 patients were categorised as having "true infections," the section on antimicrobial regimen changes refers to 124 patients. This discrepancy needs clarification.
- Lines 226-229: Since *Candida albicans* cannot be regarded as a pathogen, "prophylactic/empirical therapy" should not be guided by these findings.
- Lines 265-266: Pathogen-directed therapy should not be claimed unless results are compared with a gold standard.
- Lines 311-314: Methodology is unclear - how exactly was sputum collected (via nasal aspiration, ET, etc.)?
- Lines 316-317: Which sample was used for beta-D-glucan testing? Which kits were used for GM and BDG assays?
- General: At several places, organism names should be italicised.

Reviewer #3 (Comments for the Author):

Peer review of the article "Targeted Next-Generation Sequencing of Bronchoalveolar Lavage Fluid for Pathogen Detection in Connective Tissue Disease-Related Pulmonary Infections" (control no. Spectrum02550-25)

- Study Design and Generalizability: The retrospective, single-center design limits the external validity of the findings and may introduce selection bias. This should be more clearly acknowledged in the discussion.
- Thresholds for Pathogen Positivity: The manuscript notes the absence of standardized thresholds for defining pathogen positivity in tNGS. However, this has important implications for reproducibility and clinical decision-making. A more detailed discussion of how the authors managed this limitation would enhance the clarity and utility of the study.
- Impact of Immunosuppressive Agents: Although the influence of immunosuppressive therapies is mentioned, the study does not analyze how different regimens might affect infection risk or pathogen distribution. Including such an analysis, or at least discussing its absence, would strengthen the interpretation of results.
- Long-Term Outcomes: The study does not assess the impact of tNGS-guided treatment on longer-term clinical outcomes such as infection recurrence, survival, or mortality. Addressing this limitation or suggesting it as an avenue for future research would improve the manuscript.
- Statistical Precision: Some odds ratios (e.g., for CKD) are accompanied by wide confidence intervals, indicating imprecision. This limitation should be explicitly acknowledged in the results and discussion.
- The manuscript is generally clear, but several minor grammatical and stylistic errors remain. Careful language editing would improve readability without altering the scientific content.

Peer review of the article **Targeted Next-Generation Sequencing of Bronchoalveolar Lavage Fluid for Pathogen Detection in Connective Tissue Disease–Related Pulmonary Infections**" (control no. Spectrum02550-25)

- **Study Design and Generalizability:** The retrospective, single-center design limits the external validity of the findings and may introduce selection bias. This should be more clearly acknowledged in the discussion.
- **Thresholds for Pathogen Positivity:** The manuscript notes the absence of standardized thresholds for defining pathogen positivity in tNGS. However, this has important implications for reproducibility and clinical decision-making. A more detailed discussion of how the authors managed this limitation would enhance the clarity and utility of the study.
- **Impact of Immunosuppressive Agents:** Although the influence of immunosuppressive therapies is mentioned, the study does not analyze how different regimens might affect infection risk or pathogen distribution. Including such an analysis, or at least discussing its absence, would strengthen the interpretation of results.
- **Long-Term Outcomes:** The study does not assess the impact of tNGS-guided treatment on longer-term clinical outcomes such as infection recurrence, survival, or mortality. Addressing this limitation or suggesting it as an avenue for future research would improve the manuscript.
- **Statistical Precision:** Some odds ratios (e.g., for CKD) are accompanied by wide confidence intervals, indicating imprecision. This limitation should be explicitly acknowledged in the results and discussion.
- The manuscript is generally clear, but several minor grammatical and stylistic errors remain. Careful language editing would improve readability without altering the scientific content.

Dear Editors and Reviewers:

We truly appreciate all the constructive comments and suggestions on the manuscript (Manuscript Number: Spectrum02550-25), title: Targeted Next-Generation Sequencing of Bronchoalveolar Lavage Fluid for Pathogen Detection in Connective Tissue Disease-Related Pulmonary Infections.

Thank you very much for your effort in handling our manuscript. We greatly appreciate the thoughtful and constructive comments from the editor and the reviewers. Enclosed please find our point-by-point responses to the comments. We hope that the revised manuscript will satisfy the reviewers. Thanks again for your timely consideration of our manuscript.

Editor's comments:

Lines 89-90: "CMTs, including sputum, blood, or bronchoalveolar lavage fluid (BALF) cultures and PCR, are widely used for diagnosing lung infections": There are no references to support these statements. More references should be cited, with these (PMID: 39744807; PMID: 40137747; PMID: 33069878) as examples (citing is optional).

Response and Revision: Thank you for your review and constructive suggestion. This is a significant recommendation. As advised, we have carefully studied the three recommended references and have added those in reference as [11, 12, 13] in revised manuscript. These additions have enriched our research background and enhanced the rigor of our study. The detailed modifications were marked with blue font in the revised manuscript.

Reviewer #2 (Comments for the Author):

The manuscript is vague and difficult to follow. The study design lacks clarity, with poorly defined cohorts and the absence of appropriate controls. It is also unclear how and when the organisms were classified as pathogenic. The authors should specify whether a cut-off for tNGS was defined to consider an organism pathogenic. Furthermore, the platform used for tNGS has not been mentioned, which is an important methodological detail that must be included.

Response and Revision: Thank you for these critical comments regarding the clarity of our manuscript's methodology. We have revised the manuscript to address these points directly and improve the overall clarity of the study design, cohort definitions, and analytical methods. Below is our point-by-point response.

1. Study Design, Cohorts, and Controls

Patients with connective tissue disease were eligible if they were managed in the outpatient clinic or inpatient ward of the Department of Rheumatology and Immunology at Guangdong Second Provincial General Hospital between October 2022 and May 2025, underwent fiber-optic

bronchoscopy with bronchoalveolar lavage (BAL), and had BAL fluid submitted for tNGS. The comparison between tNGS and conventional microbiological tests (CMTs) was conducted within this single cohort. Specifically, for each patient sample, the results from tNGS performed on the BALF sample were directly compared to the results from the CMTs (which included culture, smear, etc.) . This within-sample comparative design serves as the internal control for evaluating the diagnostic performance of tNGS against the standard CMTs.

2. Classification of Organisms as Pathogenic, tNGS Cut-off and tNGS Platform

Thank you for your comments. This is a crucial point. Firstly, when selecting a tNGS testing service provider, we comprehensively considered the institution's professional qualifications, technical expertise, and industry reputation. The specimens for this study were tested by Guangzhou KingMed Diagnostics Group Co., Ltd., a pioneer and leading enterprise in China's third-party medical testing industry, which possesses significant professional advantages and market recognition in the etiological diagnosis of infectious diseases. This ensures the credibility of increasing the positive pathogen threshold in the test report. Additionally, while tNGS can detect multiple pathogens simultaneously, accurately identifying the true causative pathogen from the detected microorganisms and distinguishing whether a pathogen is in an infectious state, a colonizing state, or a result of environmental contamination is highly challenging. As you rightly pointed out, this represents a major limitation of tNGS. Following the Chinese Expert Consensus on the Application of Targeted Next-Generation Sequencing in Infectious Diseases (APA: National Center for Infectious Diseases Clinical Medical Research, Shenzhen Medical Association Infectious Diseases Professional Committee, & Shenzhen Medical Quality Control Center for Infectious Diseases. (2024). Baxiang gaotongliang cedu jishu yingyong yu ganranxing jibing zhuanjia gongshi [Expert consensus on the application of targeted high-throughput sequencing technology in infectious diseases]. *Zhonghua Linchuang Ganranbing Zazhi* [Chinese Journal of Clinical Infectious Diseases] (6), 401-412.), we interpreted tNGS results—whether positive or negative—in conjunction with the patient's clinical condition, other laboratory findings, and the pathogen's inherent pathogenicity to determine if it is the causative microorganism. When necessary, a comprehensive interpretation incorporating literature review or a multidisciplinary consultation involving infectious disease specialists, pathogenic microbiology teams, and relevant clinical departments was conducted. The detailed modifications were highlighted in blue in the revised manuscript:

On page 6, lines 258-263, "In our study, we interpreted tNGS results—whether positive or negative—in conjunction with the patient's clinical condition, other laboratory findings, and the pathogen's inherent pathogenicity to determine if it is the causative microorganism. When necessary, a comprehensive interpretation incorporating literature review or a multidisciplinary consultation involving infectious disease specialists, pathogenic microbiology teams, and relevant clinical departments was conducted."

On page 9, lines 352-370, "the "three-step comprehensive judgment" as follows:

Technical positivity

Detection limit: Sequences with ≥ 100 copies/mL are reliably detected.

Reporting symbol: A "(+)" marked in the result column indicates technical positivity.

Etiological verification

Common pathogens: Need to be re-verified using in vitro diagnostic kits for the same type of

specimen or Sanger sequencing (first-generation sequencing).

Uncommon/rare pathogens: If routine tests are already positive, they can be directly recognized; otherwise, literature evidence supporting their pulmonary pathogenicity is required.

Clinical judgment

A result can be finally confirmed as clinically positive only if at least one of the following conditions is met:

Consistent with traditional etiological results (culture, antigen/nucleic acid testing, etc.);

Consistent with the patient's symptoms, imaging findings, and treatment response;

Negative by traditional methods, but targeted medication is effective, or there is sufficient literature supporting its pathogenicity.

Whether a pathogen can be identified as the responsible one must be finally determined by clinicians based on the above composite criteria."

We believe these revisions have significantly improved the clarity and rigor of the methodological description. The changes are clearly marked in the revised manuscript, and we are grateful for the opportunity to enhance our work.

Specific comments requiring clarification:

• **Lines 71-74:** Is aspiration pneumonitis truly an extra-pulmonary manifestation? How is it linked to the digestive tract and the heart?

Response and Revision: Thank you for your professional proposal. I apologize for the logical expression error in my previous English writing. Aspiration pneumonia is not an extrapulmonary manifestation; rather, it can occur as a secondary complication following digestive dysfunction, which itself is an extrapulmonary manifestation of CTDs.

The detailed modifications were highlighted in blue in the revised manuscript:

On page 2, lines 71-78, "These events may also induce extrapulmonary complications specific to the disease, such as gastrointestinal or cardiac pathologies, and further trigger related acute respiratory events. For instance, the unique pathophysiological mechanisms in patients with scleroderma and dermatomyositis-polymyositis (DM-PM) can severely impair swallowing function, leading to aspiration pneumonia. Meanwhile, chronic inflammation associated with CTDs contributes to both atherosclerotic and non-atherosclerotic processes, which can subsequently result in acute respiratory events like pericarditis-induced dyspnea."

• **Lines 77-80:** Complications appear to be repeated.

Response and Revision: Thank you for your comment. We agree that the original phrasing could be more concise and precise. The detailed modifications were marked with blue font in the revised manuscript.

On page 2, lines 78-80, "CTDs are frequently complicated by severe pulmonary infections that are the leading cause of acute exacerbation, prolonged hospitalization, admission to the ICU and death."

• **Line 82:** The mention of culture and serodiagnostics for *Pneumocystis jirovecii* is inappropriate,

as this organism is not a good example for culture-based methods.

Response and Revision: We sincerely thank you for this insightful and correct comment. We agree that using *Pneumocystis jirovecii* as an example for the poor sensitivity of culture methods was not appropriate, as it is not routinely cultured in clinical practice. Our intention was to highlight the limitations of conventional microbiological tests (CMTs) for diagnosing challenging pathogens. We have deleted the original wording and revised the sentence to read:

On page 2, lines 80-82, "Unfortunately, conventional microbiological tests (CMTs) exhibit unacceptably low sensitivity (e.g., $\leq 30\%$ for *Pneumocystis jirovecii*), leading to frequent missed diagnoses and delayed targeted therapy."

• **Line 87:** Please clarify whether these are pulmonary opportunistic infections.

Response and Revision: Thank you for your comment. I apologize for the logical expression error in my previous English writing. We have revised the sentence in the manuscript and the detailed modifications were marked with blue font in the revised manuscript.

On page 3, lines 85-90, "The primary pathogens causing infections in PM/DM patients include *Staphylococcus aureus*, *Klebsiella*, *Escherichia coli*, *Mycobacterium*, and *Salmonella*. The most common infections were aspiration pneumonia (21.1%) and opportunistic infections (18.4%). Among the opportunistic infections, *Mycobacterium* was the most prevalent, accounting for half of all opportunistic infection episodes, followed by cytomegalovirus (CMV) infection."

• **Line 138:** *Candida albicans* should not be considered pathogenic flora in BALF.

Response and Revision: Thank you for raising this important point regarding the interpretation of *Candida albicans* detection in our study. We fully agree with the reviewer that *C. albicans* is a classic opportunistic pathogen. It commonly colonizes mucosal surfaces such as the oral cavity, upper respiratory tract, intestines, and vagina in healthy individuals, typically existing in a commensal state without causing disease. It becomes pathogenic primarily when the host's immune function is compromised or when the normal microbial flora is disrupted, allowing it to proliferate excessively and invade tissues.

The CTD patient had been treated with glucocorticoids, mycophenolate mofetil, cyclosporine, and cyclophosphamide. Both the underlying disease and these immunosuppressive agents severely compromised host immunity, predisposing the patient to opportunistic fungal infections. Moreover, the marked clinical response to antifungal therapy supports the conclusion that this patient had a concomitant *Candida* infection. Therefore, in compiling the final results, we only included *Candida* species that were conclusively determined to be pathogenic based on the above rigorous criteria. We believe this approach is appropriate and ensures that the pathogens we report are clinically relevant, thereby avoiding the overinterpretation of colonization or contamination. We appreciate the opportunity to clarify this methodological detail.

• **Line 166:** While only 109 patients were categorised as having "true infections," the section on antimicrobial regimen changes refers to 124 patients. This discrepancy needs clarification.

Response and Revision: Thank you very much for the reminder. Of the 124 specimens, tNGS detected pathogens in 105, whereas conventional microbiological tests (CMTs) were positive in only 42. After comprehensive review of clinical, radiological and therapeutic information, 109 episodes were classified as true infection. Because connective-tissue diseases can themselves

cause fever and pulmonary infiltrates, many patients without infection received empirical antibiotics while the possibility of infection was still under evaluation; these antibiotics were stopped once infection was confidently excluded.

- **Lines 226-229:** Since *Candida albicans* cannot be regarded as a pathogen, "prophylactic/empirical therapy" should not be guided by these findings.

Response and Revision: Thank you very much for the advice. The CTD patient had been treated with glucocorticoids, mycophenolate mofetil, cyclosporine, and cyclophosphamide. Both the underlying disease and these immunosuppressive agents severely compromised host immunity, predisposing the patient to opportunistic fungal infections. Moreover, the marked clinical response to antifungal therapy supports the conclusion that this patient had a concomitant *Candida* infection.

- **Lines 265-266:** Pathogen-directed therapy should not be claimed unless results are compared with a gold standard.

Response and Revision: Thank you very much for the helpful suggestion. We have deleted the original wording and revised the sentence to read:

“By enabling rapid pathogen detection, BALF-tNGS has the potential to improve outcomes, reduce antimicrobial misuse, and mitigate the burden of infections in this high-risk population.”

- **Lines 311-314:** Methodology is unclear - how exactly was sputum collected (via nasal aspiration, ET, etc.)?

Response and Revision: Thank you for your comment regarding the clarification of our sputum collection methodology. We apologize for the lack of detail in the original description. We have revised the sentence in the manuscript and the detailed modifications were marked with blue font in the revised manuscript.

On page 8, lines 338-340, "Before the first dose of antibiotics, sputum was collected by natural expectoration, sputum catheter aspiration, and bronchoscopic collection and sent for bacterial culture."

- **Lines 316-317:** Which sample was used for beta-D-glucan testing? Which kits were used for GM and BDG assays?

Response and Revision: Thank you for the constructive comments. The GM and BDG assays were performed using the Fungal (1-3)- β -D-Glucan Detection Kit (Chromogenic Method) and the Aspergillus Galactomannan Detection Kit (Chemiluminescence Method) from China's Xinuo Company. The BALF was submitted for sputum smear, bacterial culture, (1,3)- β -D-glucan test, Galactomannan test, and tNGS.

The information has been added to the manuscript.

- **General:** At several places, organism names should be italicised.

Response and Revision: Thank you for your important reminder regarding the correct formatting of organism names. We completely agree that adhering to standard microbiological nomenclature is crucial for scientific precision. We have thoroughly reviewed the entire manuscript and ensured that all scientific names of organisms at the genus and species level are now consistently italicized

throughout the text, including in the abstract, main text, figures, and tables. We appreciate this valuable feedback, which has helped us improve the manuscript's adherence to formal scientific writing conventions. The changes have been marked with blue font in the revised manuscript.

Reviewer #3 (Comments for the Author):

Peer review of the article Targeted Next-Generation Sequencing of Bronchoalveolar Lavage Fluid for Pathogen Detection in Connective Tissue Disease-Related Pulmonary Infections" (control no. Spectrum02550-25)

• **Study Design and Generalizability:** The retrospective, single-center design limits the external validity of the findings and may introduce selection bias. This should be more clearly acknowledged in the discussion.

Response and Revision: Thank you for raising this important question. We have further supplemented the discussion and conclusion sections of the original manuscript to address the limitations of this single-center, retrospective study and to add future research directions. The detailed modifications were highlighted in blue in the revised manuscript.

On page 6, lines 252-255, "Our study has several limitations. First, it is a single-center retrospective analysis, which may limit generalizability due to potential biases in patient selection and local pathogen epidemiology. Future research should aim to validate these findings through multi-center, prospective cohort designs and include broader patient populations to enhance external validity ."

On page 7, lines 280-281, "Future multicenter prospective studies are needed to validate these findings and establish standardized protocols for tNGS implementation in CTD care."

• **Thresholds for Pathogen Positivity:** The manuscript notes the absence of standardized thresholds for defining pathogen positivity in tNGS. However, this has important implications for reproducibility and clinical decision-making. A more detailed discussion of how the authors managed this limitation would enhance the clarity and utility of the study.

Response and Revision: Thank you for your comments. Firstly, when selecting a tNGS testing service provider, we comprehensively considered the institution's professional qualifications, technical expertise, and industry reputation. The specimens for this study were tested by Guangzhou KingMed Diagnostics Group Co., Ltd., a pioneer and leading enterprise in China's third-party medical testing industry, which possesses significant professional advantages and market recognition in the etiological diagnosis of infectious diseases. This ensures the credibility of increasing the positive pathogen threshold in the test report. Additionally, while tNGS can detect multiple pathogens simultaneously, accurately identifying the true causative pathogen from the detected microorganisms and distinguishing whether a pathogen is in an infectious state, a colonizing state, or a result of environmental contamination is highly challenging. As you rightly pointed out, this represents a major limitation of tNGS. Following the Chinese Expert Consensus on the Application of Targeted Next-Generation Sequencing in Infectious Diseases (APA: National Center for Infectious Diseases Clinical Medical Research, Shenzhen Medical Association Infectious Diseases Professional Committee, & Shenzhen Medical Quality Control Center for

Infectious Diseases. (2024). Baxiang gaotongliang cedu jishu yingyong yu ganranxing jibing zhuanjia gongshi[Expert consensus on the application of targeted high-throughput sequencing technology in infectious diseases]. *Zhonghua Linchuang Ganranbing Zazhi*[Chinese Journal of Clinical Infectious Diseases](6), 401-412.), we interpreted tNGS results—whether positive or negative—in conjunction with the patient's clinical condition, other laboratory findings, and the pathogen's inherent pathogenicity to determine if it is the causative microorganism. When necessary, a comprehensive interpretation incorporating literature review or a multidisciplinary consultation involving infectious disease specialists, pathogenic microbiology teams, and relevant clinical departments was conducted. The detailed modifications were highlighted in blue in the revised manuscript:

On page 6-7, lines 258-263, "In our study, we interpreted tNGS results—whether positive or negative—in conjunction with the patient's clinical condition, other laboratory findings, and the pathogen's inherent pathogenicity to determine if it is the causative microorganism. When necessary, a comprehensive interpretation incorporating literature review or a multidisciplinary consultation involving infectious disease specialists, pathogenic microbiology teams, and relevant clinical departments was conducted."

On page 9, lines 352-370, "the "three-step comprehensive judgment" as follows:

Technical positivity

Detection limit: Sequences with ≥ 100 copies/mL are reliably detected.

Reporting symbol: A "(+)" marked in the result column indicates technical positivity.

Etiological verification

Common pathogens: Need to be re-verified using in vitro diagnostic kits for the same type of specimen or Sanger sequencing (first-generation sequencing).

Uncommon/rare pathogens: If routine tests are already positive, they can be directly recognized; otherwise, literature evidence supporting their pulmonary pathogenicity is required.

Clinical judgment

A result can be finally confirmed as clinically positive only if at least one of the following conditions is met:

Consistent with traditional etiological results (culture, antigen/nucleic acid testing, etc.);

Consistent with the patient's symptoms, imaging findings, and treatment response;

Negative by traditional methods, but targeted medication is effective, or there is sufficient literature supporting its pathogenicity.

Whether a pathogen can be identified as the responsible one must be finally determined by clinicians based on the above composite criteria."

• **Impact of Immunosuppressive Agents:** Although the influence of immunosuppressive therapies is mentioned, the study does not analyze how different regimens might affect infection risk or pathogen distribution. Including such an analysis, or at least discussing its absence, would strengthen the interpretation of results.

Response and Revision: Thank you for raising this critical point. We completely agree that the impact of specific immunosuppressive regimens is a crucial factor. Unfortunately, due to the retrospective nature of our study and the relatively small sample size within each specific

immunosuppressive regimen subgroup (e.g., only a small number of patients received biologic agents), a robust statistical comparison of infection risks across different regimens was not feasible. More importantly, following your suggestion, we have now explicitly acknowledged this as a limitation of our study . We have also incorporated a discussion on the established literature regarding the infection risks of different drug classes to provide context and strengthen the interpretation(PMID: 38251591, PMID: 33597206). The detailed modifications were highlighted in blue in the revised manuscript:

On page 7, lines 263-277, "Third, due to the retrospective nature of our study and the relatively small sample size within each specific immunosuppressive regimen subgroup (e.g., only a small number of patients received biologic agents), a robust statistical comparison of infection risks across different regimens was not feasible. We did not systematically assess the impact of specific immunosuppressive agents (e.g., rituximab, cyclophosphamide) on infection risk, which warrants further investigation. Literature indicates that immunosuppressive therapy significantly increases the risk of infection. In patients with rheumatic diseases receiving immunosuppressants (such as rituximab, belimumab, and traditional immunosuppressive agents), the risk of severe infection is less associated with the specific type of drug and more dependent on the patient's underlying condition. Immunosuppressive therapy for rheumatic diseases significantly alters the spectrum of pathogens, predominantly leading to opportunistic infections (e.g., PJP, tuberculosis) and respiratory pathogens (e.g., pneumococcus, viruses). The distinct immunodeficiency profiles of different rheumatic diseases (such as systemic lupus erythematosus and rheumatoid arthritis) may result in specific pathogen susceptibilities; for instance, SLE patients are more susceptible to tuberculosis infection [27, 28]. "

- **Long-Term Outcomes:** The study does not assess the impact of tNGS-guided treatment on longer-term clinical outcomes such as infection recurrence, survival, or mortality. Addressing this limitation or suggesting it as an avenue for future research would improve the manuscript.

Response and Revision: Thank you for your comment. We completely agree that evaluating the impact of tNGS-guided therapy on longer-term clinical outcomes is of great clinical significance. Our study, with its primary focus on the diagnostic performance and short-term therapeutic adjustment based on tNGS, did not extend to assessing long-term endpoints such as infection recurrence or mortality, which is indeed a limitation. In response to this valuable suggestion, we have now added a future perspectives to explicitly acknowledge this point and propose directions for future research. The detailed modifications were highlighted in blue in the revised manuscript:

On page 7, lines 277-281, "Finally, the study primarily assessed the short-term impact of tNGS on empirical antimicrobial therapy adjustment. long-term outcomes (e.g., recurrence rates, mortality) were not evaluated, limiting our ability to correlate tNGS-guided therapy with sustained clinical benefit. Future multicenter prospective studies are needed to validate these findings and establish standardized protocols for tNGS implementation in CTD care."

- **Statistical Precision:** Some odds ratios (e.g., for CKD) are accompanied by wide confidence intervals, indicating imprecision. This limitation should be explicitly acknowledged in the results and discussion.

Response and Revision: We appreciate this observation. In response, we have added the

following sentence to the Discussion:

“However, some odds ratios (e.g., for CKD) are accompanied by wide confidence intervals, reflecting the small subgroup sample and the low event count; consequently, these estimates are imprecise and should be interpreted cautiously, and no definitive quantitative risk estimate can be derived from the present data.”

- The manuscript is generally clear, but several minor grammatical and stylistic errors remain. Careful language editing would improve readability without altering the scientific content.

Response and Revision: Thank you again for your constructive comments.

Re: Spectrum02550-25R1 (**Targeted Next-Generation Sequencing of Bronchoalveolar Lavage Fluid for Pathogen Detection in Connective Tissue Disease-Related Pulmonary Infections**)

Dear Prof. Zhengping Huang:

Your manuscript has been accepted, and I am forwarding it to the ASM production staff for publication. Your paper will first be checked to make sure all elements meet the technical requirements. ASM staff will contact you if anything needs to be revised before copyediting and production can begin. Otherwise, you will be notified when your proofs are ready to be viewed.

Sincerely,
Benjamin Liu
Editor
Microbiology Spectrum